# Simplified Numerical Model for Determining Load-Bearing Capacity of Steel-Wire Ropes

**DOI:** 10.3390/ma16103756

**Published:** 2023-05-16

**Authors:** Juraj Hroncek, Pavel Marsalek, David Rybansky, Martin Sotola, Lukas Drahorad, Michal Lesnak, Martin Fusek

**Affiliations:** 1Department of Applied Mechanics, Faculty of Mechanical Engineering, VŠB-Technical University of Ostrava, 17. listopadu 2172/15, 708 00 Ostrava, Czech Republic; 2Department of Work and Process Safety, Faculty of Safety Engineering, VŠB-Technical University of Ostrava, 17. listopadu 2172/15, 708 00 Ostrava, Czech Republic; 3Nanotechnology Centre, VŠB-Technical University of Ostrava, 17. listopadu 2172/15, 708 00 Ostrava, Czech Republic

**Keywords:** load-bearing capacity, steel rope, modeling, tensile test, plastic strain, failure, finite element method

## Abstract

Steel-wire rope is a mechanical component that has versatile uses and on which human lives depend. One of the basic parameters that serve to describe the rope is its load-bearing capacity. The static load-bearing capacity is a mechanical property characterized by the limit static force that the rope is able to endure before it breaks. This value depends mainly on the cross-section and the material of the rope. The load-bearing capacity of the entire rope is obtained in tensile experimental tests. This method is expensive and sometimes unavailable due to the load limit of testing machines. At present, another common method uses numerical modeling to simulate an experimental test and evaluates the load-bearing capacity. The finite element method is used to describe the numerical model. The general procedure for solving engineering tasks of load-bearing capacity is by using the volume (3D) elements of a finite element mesh. The computational complexity of such a non-linear task is high. Due to the usability of the method and its implementation in practice, it is necessary to simplify the model and reduce the calculation time. Therefore, this article deals with the creation of a static numerical model which can evaluate the load-bearing capacity of steel ropes in a short time without compromising accuracy. The proposed model describes wires using beam elements instead of volume elements. The output of modeling is the response of each rope to its displacement and the evaluation of plastic strains in the ropes at selected load levels. In this article, a simplified numerical model is designed and applied to two constructions of steel ropes, namely the single strand rope 1 × 37 and multi-strand rope 6 × 7-WSC.

## 1. Introduction

The need to advance the production of steel ropes is closely linked to the enormously rapid development of construction, engineering, and industry in general in recent decades. New materials and technologies have enabled productivity growth in line with progress trends. Today’s steel ropes offer a versatile and economical solution to almost all problems associated with the transmission of mechanical energy by traction.

A steel-wire rope is a mechanical device that has many moving parts working in tandem to help support and move an object or load. In the lifting equipment industry, a steel-wire rope is attached to a crane or hoist and is equipped with swivels, stirrups, or hooks. It can also be used to raise and lower elevators or as a means of supporting suspension bridges or towers. Steel-wire rope is the preferred lifting device for many reasons. Its unique design consists of several steel wires that form individual strands stored in a spiral around the core. The construction of the rope must meet its basic requirement—to achieve a high load-bearing capacity with a relatively small diameter and, at the same time, have low mass per length and sufficient flexibility. Different materials and wire and strand structures will provide various benefits for a particular application. The advantages include, for example, strength, flexibility, resistance to rotation, and resistance to abrasion, fatigue, and corrosion [1,2,3].

Wire rope, like any machine part, requires proper selection, installation, and maintenance. If one of these is performed incorrectly, the service life of the rope will be dramatically shortened. In the article [4], CraneTech addressed possible causes of steel-wire rope failure. The most common causes include abrasive failure, corrosion failure, core protrusion, fatigue failure, rope breakage failure, or rope overload failure. Rope overload is a common cause of rope failure. With this type of damage, caused by excessive tension or excessive impact load, the wires or the rope itself break. There is typically a narrowing of the broken wire ends, a so-called cramp. Mouradi et al. [5] investigated the sudden failure of steel ropes due to broken wires. Ropes that are subjected to high tensile stress and negligible bending stress tend to fail from the inside. This phenomenon is described by Verreet [6] and also by Morelli [7]. From a safety point of view, this is a very dangerous type of rope failure as, in most cases, it is impossible to predict. It follows that the load-bearing capacity of the rope is an extremely important factor in the relevant rope construction.

Currently, three methods are commonly used for determining the load-bearing capacity and overall physical properties of steel ropes. These include an experimental approach, an approach using the empirical relationships of individual analytical methods, and, currently, the most progressive modeling and simulation using modern computer technology. The principles of tensile tests of a single wire rope strand are described by Utting and Jones [8]. Other authors who describe the experimental testing of steel ropes are Yusuf Aytaç Onur [9] and Gina Diana Musca [10]. However, laboratory testing of ropes is not always possible due to the load limit of testing machines. Currently, destructive experimental tests of steel ropes are not the only type of test available. Thanks to non-destructive tests, it is possible to determine the properties, condition, or damage of the rope without destroying it. Zhou et al. describe the most widely used methods for the non-destructive testing of steel ropes [11].

Prior to the advent of the numerical modeling method, many authors tried to describe the theoretical behavior of steel ropes under their load. Machida and Durelli [12] investigated the theoretical and experimental behavior of oversized plastic and wire single-fiber ropes. Costello et al. [13,14] used a basic approach to model the elastic response of six-strand ropes without an inner core. Velinsky et al. [15] considered Poisson effects on the response of a seven-strand wire-core rope and the more complex structures of strands and ropes. They linearized non-linear equations of equilibrium and, in a later article, Velinsky [16] showed that predictions from linear and non-linear theories were almost identical in the load range in which most wire ropes are used. Utting and Jones [17] introduced a new mathematical model of rope response, which takes into account the friction between individual wires, the effects of the Poisson ratio, and the flattening of individual wires. In the following years, several analytical models were developed that were able to predict the mechanical behavior of steel ropes subjected to axial loads based on knowledge of the material behavior, geometry, and wire construction. The study of analytical models is discussed in detail by Ghoreishi [18].

Currently, the most successful and innovative approach that can predict the mechanical behavior of steel ropes involves modeling and simulating the relevant rope loading conditions using computer technology. This approach is also used in the publication [19] to model the load-bearing capacity of a single-stranded steel rope, which this article builds on. The principles of finite element modeling are further used, for example, by Foti and di Riseto [20] and Judge et al. [21]. Many authors have constructed a numerical model of steel ropes using the volume elements of a finite element mesh. Both approaches, i.e., the use of volume and of beam mesh elements, were compared in his work by the author Bruski [22]. He discovered that the differences in modeling with the given approaches are small and the behavior of the rope in the relevant analyses is similar. He further emphasized a reduction in the complexity of the model (a reduction of up to 99.6%). This directly affects the time and complexity of the calculation and the required memory of the data station. A similar idea to simplify the modeling of the load-bearing capacity of steel ropes using beam elements is also discussed in this publication.

The basic principle of the modeling of steel rope simulation is to create a non-linear numerical model whose material, geometry, and behavior at different loads most closely resemble a real rope. The first step is to create a geometric model that has the same dimensions as a real rope. The next step is to define the material parameters of the rope, obtained from experimental tests or standards, or tables. When using the finite element method, the relevant geometry is discretized into small cells called elements that are connected by nodes. These elements are far easier to calculate than the original geometry. In mechanical applications, the primary variables are the displacements of nodes. This is calculated by determining local and global stiffness matrices (which might or might not change during solving). The next step is to define the boundary conditions (for example, attachment in the chucks during the tensile test, contacts occurring between the wires) and the loads that act on the rope in the modeled application. After the successful creation of a model whose geometry and behavior correspond to reality, the last step is the solution of the non-linear model. The method of numerical modeling is accurate, but, in some applications, it is extremely time-consuming [19].

There are several ways to speed up the calculation of the numerical analysis. This includes a possible parallelization of the given task and its effective simultaneous solution on a supercomputer or a very powerful device. This procedure is described in detail by the authors Miyamura and Yoshimura in their article [23]. The second way to speed up the calculation is to simplify the numerical model.

This article deals with the preparation and validation of a simplified numerical model for calculating the load-bearing capacity of steel ropes using the finite element method. The main goal of this article is to speed up the calculation of steel ropes without compromising accuracy due to its implementation in practice. The first analyzed rope is a 1 × 37 single-strand rope. The task was addressed in the previous publication [19], where the authors prepared a complex dynamic numerical model. The new approach uses beam elements and a static numerical model of the rope with the help of available data from experimental tensile tests published in [19]. In the second part of the article, a validated numerical model is applied to the multi-strand rope 6 × 7-WSC. The preliminary results from the article were published by the first author of this study in his diploma thesis [24].

## 2. Material and Methods

In this paper, the authors used the numerical modeling method for a description of the tensile test of two wire ropes: (i) Single-strand rope 1 × 37. (ii) Multi-strand rope 6 × 7-WSC. The description of the numerical model is divided into four parts: a geometric model, a finite element mesh, a material model, and the boundary conditions. The proposed non-linear model is calculated using Ansys Workbench 2020R2 (Ansys, Canonsburg, PA, USA). The authors of this article have many years of experience in the application of numerical methods to solve the load-bearing capacity of complex structures [25,26,27].

### 2.1. Geometric Model

The creation of a geometric model of rope construction 1 × 37 is based on the experiment and publication [19], which this article builds on. In the experimental tests, a sample of a right-hand Lang lay rope with an outer diameter *d* = 8.00 mm, a length *l* = 150 mm, and a pitch height of the helix per thread *h* = 50 mm is used. The rope consists of 37 wires laid in four layers with a construction of 1 + 6 + 12 + 18 wires in individual layers. Figure 1 shows the cross-section of the rope; the measured diameters in the individual layers are listed in Table 1 [19].

There are many ways to create the wire rope geometry, namely mathematical modeling using equations, scanning of the real rope, creating a geometric model in CAD software, and more. The mathematical modeling is covered in detail by the author Zhang et al. [28], while the scanning of a different application using CT is covered by Sejda et al. [29]. In our case, the third method of creating a geometric model was chosen.

In the SpaceClaim modeling environment of the Ansys Workbench software, a cross-section is initially created according to the wire diameters listed in Table 1 and according to Figure 1. The next step is to create a rope sample of the specified length. This is achieved by pulling the cross-sections of the individual wires along the helix with a specified pitch height around the axis of the inner layer.

In the case of the second rope, it is a construction of a multi-strand, right-hand Lang lay 6 × 7-WSC steel rope. The construction of the rope consists of two layers of steel strands. The core is made up of the same steel strand of wires as the outer layers. Experimental tests are simulated using rope samples with an outer diameter *d* = 8.00 mm, length *l* = 150 mm, and helix pitch height per 1 thread *h* = 70 mm. The rope cross-section is defined in standard EN 12385-4 from 2002 and shown in Figure 2. The rope consists of 49 steel wires of the same diameter. The diameter of the individual wires is $d_{d}$ = 0.88 mm.

The creation of a geometric model in SpaceClaim differs slightly from the process of creating a 1 × 37 rope model. There is a significant difference in the way the rope is braided, as well as, of course, in the cross-section. During production, the wires are first braided into strands, and then the strands are braided together, which produces the resulting rope. This procedure is also followed when creating the geometric model of the rope. It is a right-hand Lang lay, which means that the wires in the strands are wound in the same direction and with the same pitch height as the strands forming the entire rope.

### 2.2. Finite Element Mesh

Creating a finite element mesh is relatively simple when using beam elements. Each wire of the rope representing one beam is divided into 100 elements. The element formulation BEAM188 is used in the analyses. It is a one-dimensional line element in space. BEAM188 is the element suitable for analyzing slender to moderately thick beam structures. The element is based on the Timoshenko beam theory which includes shear-deformation effects. To describe the individual wires, a linear approximation of displacements is used. In this case, BEAM188 consists of two nodes and each node has six degrees of freedom. These include translations in the *x*-, *y*-, and *z*-axes and rotations about the *x*-, *y*-, and *z*-axes. The BEAM188 element describes the cross-section of each wire using the circular solid section type (CSOLID). For the description of CSOLID, the circumference of the wire is divided into eight integration points. The second layer of integration points is placed on an imaginary circle with a smaller diameter than the cross-section. Overall, 16 + 1 (1 in the middle of the cross-section) integration points are used to describe the cross-section. More integration points improve the accuracy slightly but may affect the convergence of the analysis [30]. The BEAM188 element has been reliably used in publications by Zhang et al. [31] and by Agabito [32] in the analysis of steel ropes.

A sensitivity analysis of the mesh impact was performed, which revealed that further reduction in the element size and the associated refinement of the mesh increased computational time, but the differences in the results were negligible. The rope 1 × 37 finite element mesh is shown in Figure 3.

The principle of creating a finite element mesh of a 6 × 7-WSC rope construction is the same as in the case of the 1 × 37 construction. The rope 6 × 7-WSC finite element mesh is shown in Figure 4 and the mesh statistics of both ropes are listed in Table 2.

### 2.3. Material Model

The mechanical properties of the wires are influenced by the production technology. In this case, it is not possible to work with tabular values of raw material parameters and, therefore, it is necessary to test the separate rope wires after production. The accurate description of the real behavior of wires using the material model is achieved for an elasto-plastic material model with bilinear isotropic hardening. All the mechanical properties listed in Table 3 are taken from the article which this paper builds on [19]. Experimental measurements on separate wires of rope 1 × 37 were performed and the yield strength $\sigma_{y}$, tangent modulus $E_{t}$, modulus of elasticity *E* and Poisson ratio μ were evaluated. The obtained mechanical properties were subsequently used in the numerical model. The same material model and mechanical properties were used for all rope wires. In the case of rope 6 × 7-WSC, the mechanical properties obtained for rope 1 × 37 were used.

### 2.4. Boundary Conditions

The boundary conditions are defined to reflect the experimental test. The effect of compressing the ends of the rope with the jaws is neglected. The first end of the rope is fixed. All displacements and rotations, except axial displacement, are fixed on the second end. The simulation is controlled by axial displacement applied on the second end. The prescribed final value of axial displacement is *u* = 7 mm.

The contact procedure using the GCGEN command is applied to implement the contact description between the individual wires. Contact element formulation CONTA177 is used. The contact algorithm uses an augmented Lagrangian formulation and isotropic Coulomb friction without dependency on temperature, time, normal pressure, sliding distance, or sliding relative velocity. To describe the wire contact, the traction-based model is used. The contact stiffness is updated for each iteration. This non-linear contact formulation is used to control the penetration of surfaces. If the surfaces do not penetrate each other, they are not in contact. If there is penetration, there is contact in the area of interest. The procedure automatically generates contact pairs with switches between parallel and crossing wires [30]. The algorithm works with friction, where the set value of the shear friction coefficient is *f* = 0.10. Different values of the shear friction coefficient *f* between the rope wires are worked with and its influence on the calculation is monitored. Gradually, the values of this coefficient from *f* = 0.10 to *f* = 0.50 are taken into account, but this change does not affect the results of the calculations. The reason is that the shear friction coefficient has a minimal effect on the immediate load-bearing capacity, but it does affect the wear of the rope.

A modified Newton–Raphson method is used to solve the problem. The solution is divided into 700 steps with the possibility of dividing up to 1E7 steps. Up to 200 iterations can be performed in one step to find an equilibrium. The tolerance of the force convergence is chosen to be 0.5%.

## 3. Results

The first task is to evaluate the dependence of the axial force of the single-strand rope 1 × 37 on its displacement and subsequent comparison with the results obtained from the experimental tensile tests. The authors also present the results obtained from the 3D dynamic model of this rope, which were published in [19]. This dependence and its comparison can be seen in Figure 5. To compare the results obtained from numerical modeling, the calculated rope displacement is shifted by $u_{nor}$ = 0.25 mm. This shift corresponds to the limitations of the wires during the tensile test simulation that arises due to the idealization of the rope model.

It is possible to observe a step-change in the rope response in the case of modeling (a simplified model) on the graph. This break in the curve is a response to the jump of the rope wires into the gaps in the individual layers during the finite element analysis. This phenomenon is influenced by the construction, the idealization of the geometric model, the chosen discretization of the model, and the number of integration points describing the cross-section. In the case of a single-strand rope, a step change with, for example, a finer finite element mesh, occurs earlier, and, at the same time, this change is smaller.

The evaluation of equivalent plastic strain is implemented in the Ansys Mechanical APDL environment (Ansys, Canonsburg, United States of America) for three levels of axial force loading (corresponds to the load-bearing capacity). They are the values of the axial forces $F_{c}$ = 45 kN, $F_{c}$ = 50 kN, and $F_{c}$ = 55 kN. First, in the simulation of the results, the distance areas $l_{c}$ = 20 mm are removed from each end of the rope due to the influence of the jaws with which the rope is attached. The results are skewed in these areas. This is followed by the creation of a cut in the center of the rope needed to draw the contours of equivalent plastic strains and stresses. Failure of the wires is assessed based on the equivalent plastic strain evaluation (3D stress combination). The following Figure 6 shows the equivalent plastic strain fields at the evaluated axial force load levels.

It is clear from these figures that the rope initially fails from the inside since it is the inner wires of the rope that are most heavily loaded and it is these wires that reach the highest values of equivalent plastic strain. The relevant values of equivalent plastic strains $\epsilon_{pl}$ are achieved at the corresponding loads. The values of the axial forces causing these deformations correspond to the load-bearing capacity of the rope. Table 4 shows the maximum values of equivalent plastic strain $\epsilon_{pl}$ at the monitored axial force load levels.

Steel ropes break at different levels of force, which achieves a large dispersion. It is possible to see in Figure 5 that it is never an immediate rupture of the entire rope, but individual wires of the rope gradually crack. If we allow an error of up to 2%, the failure of the first wire almost always corresponds to the experimentally determined load-bearing capacity of the rope (see Table 5). For this reason, it is assumed from the numerical results that the failure of the first wire corresponds to its load-bearing capacity.

Based on experimental measurements, it is possible to observe that rope 1 × 37 starts to fail at a load of $F_{c}$ ≅ 50 kN. The equivalent plastic strain $\epsilon_{pl}$ = 3.93 × 10${}^{-2}$ belonging to the given load is assumed as the limit value for the relevant material model.

The second task is to evaluate the dependence of the axial force of the multi-strand rope 6 × 7-WSC on its displacement. In Figure 7, we can observe the plot of the curve in comparison with the rope 1 × 37.

In the case of a multi-strand rope, discretization does not have a significant effect on the step-change phenomenon. Due to the different construction of the rope, this effect is minimal and can be neglected.

Again, the evaluation of equivalent plastic strain fields is performed in selected levels of axial force loading (see Figure 8). These are the values ofthe axial forces $F_{c}$ = 35 kN, $F_{c}$ = 40 kN, and $F_{c}$ = 45 kN.

Table 6 shows the maximum values of equivalent plastic strain $\epsilon_{pl}$ at the monitored axial force load levels.

Due to the use of the same material model in the case of the 6 × 7-WSC rope as well as the 1 × 37 rope, it can be assumed that the 6 × 7-WSC rope starts to fail at the same equivalent plastic strain limit value $\epsilon_{pl}$ = 3.93 × 10${}^{-2}$. Based on this value of equivalent plastic strain, the load-bearing capacity of this rope is in the range $F_{c}$ = 35–40 kN.

## 4. Discussion

This article builds on the scientific work of Lesnak [19]. Lesnak published the 3D dynamic model solved using LS-DYNA R7.1 (LSTC, Livermore, United States of America) on the workstation Intel Core i5-3350P, 4 CPU, 16 GB RAM, 120 GB SSD. The 3D model uses hexahedral elements with full integration (860,000 nodes, 630,000 elements). It contains criteria for the failure of wires based on exceeding the equivalent plastic strain. The maximal equivalent plastic strain obtained from the 3D dynamic model is lower due to the implementation of a relatively high loading speed *v* = 5 m/s. Failure here already occurs at a smaller displacement, see Figure 5. The results obtained from the 3D dynamic model require the use of a low-pass filter to eliminate the effect of high frequencies on the calculated response, which can affect the results. The model uses an explicit time integration and the central difference method is applied. The model includes a stability criterion defined by the maximum size of the time step Δ*t* = 2.8 × 10${}^{-5}$ ms. Because of this, the total calculation time is $t_{c}$ = 123.3 h. The time required (of the order of several days, depending on the numerical model and workstation parameters) for their calculation and real non-usability in practice offered space for further research and led to the search for a more useful way of modeling the load-bearing capacity of steel ropes.

It should be noted that there was a considerable simplification in the modeling when working only with the variant that each layer of wires had the same material model or where the lubrication of the rope was not taken into account. Another simplification was the fact that the process of rope braiding and its effect on the rope was not taken into account. This is because, when the rope is braided, initial internal stress occurs and an initial equivalent plastic strain begins to occur in some individual wires, which was reported by Song et al. [33]. In addition, the equivalent plastic strain occurring inside the rope is difficult to capture during the experimental measurement. It is, therefore, difficult to compare it with the results of numerical analysis. This can be achieved in the case of testing only the individual wires of the rope.

Idealization certainly occurs even when the rope comes from the factory in an ideal condition without any defects. The ideal model was also worked on, so it was possible to simplify the model using beam elements of the finite element mesh. In the case of taking rope defects into account, it would be necessary to work with the volume elements of the finite element mesh. The effect of defects on the load-bearing capacity of steel ropes could be the subject of further research. The effect of defects on the mechanical properties of steel ropes is experimentally described in the publications [34,35]. The numerical approach to solving rope defects is applied in the publications [36,37].

Finally, a multi-strand steel rope construction 6 × 7-FC was modeled, which, instead of a wire core formed by the same strand, has a fiber core. Initially, the geometric model was created with empty space instead of the core, but this led to incorrect results. Furthermore, the geometry of the fiber core was replaced by a cylinder with a different material model than the rest of the rope. In this case, there is a problem in the contact of two different materials with a large difference in the modulus of elasticity *E*. As this is a complex task, this type of rope has not been examined and will be the subject of further study.

The stress–strain curve can be evaluated for the tensile test of one wire, where we can assume a uniaxial stress state. In the case of a tensile test of the entire rope, it is not possible to assume a uniaxial stress state in each wire. From the point of view of the issue of strain determination, it is necessary to emphasize that the wires have different lengths in the individual layers. The issue is also addressed in the article by Xiang et al. [38]. If we wanted to compare the tension obtained from the numerical simulation, it would be necessary to use some more sophisticated method than the tensile test, for example, the X-ray method, but this was not carried out. This method is described in the article by Morelli et al. [7]. Actual values of limit-equivalent plastic strain and ultimate stress can be obtained from extensive experimental tensile tests of individual wires. The individual values have a large dispersion and may also differ based on the rope manufacturer. The authors limit themselves to the evaluation of the equivalent plastic strain at chosen levels and present the possibilities of evaluating the load-bearing capacity. In subsequent works, the authors will focus on refining the material model of individual wires. Specifically, each layer of wires will have different mechanical properties. Furthermore, more complex elasto-plastic material models than a bilinear model with isotropic hardening will be used.

## 5. Conclusions

With the arrival of new types of rope structures and improved production technologies came the need to determine the true load-bearing capacity value by means other than experimental tests. The main goal of this work was to reduce the time required for the calculation and the associated streamlining of the assessment of the load-bearing capacity of steel ropes using computer modeling. An order of magnitude reduction in the calculation time from several days to the value of the total calculation time $t_{c}$ = 2.7 h was achieved with simultaneous modeling equivalent to the real behavior of the rope (workstation Intel Core i5-4310U, 2 cores, 8 GB RAM, 500 GB SSD). This simplification was achieved by deploying the beam elements of the finite element mesh, instead of the volume elements. Although a completely different approach was chosen, it leads to the same goal. The main advantage of this procedure is the achievement of a simpler numerical model, thanks to which it is possible to obtain an order of magnitude reduction in the computational and time-consuming complexity of the task without losing the accuracy of the calculation.

Two construction types of steel ropes were dealt with in the article. A single-strand rope construction 1 × 37 and a multi-strand rope construction 6 × 7-WSC were used. In the case of the single-strand rope, available experimental data were used. During modeling, the same geometric model and test method were simulated as in the case of the experiment. Figure 5 shows that similar results of rope response to its load in the tensile test were achieved as in the experiment. After successful verification of the numerical model in the case of a single-strand rope, this model was applied in the calculation of the load-bearing capacity of a multi-strand rope. Another important aspect was the evaluation of the load-bearing capacity of the ropes in the selected levels of load values and the corresponding values of equivalent plastic strains in the cross-sections of the ropes (see Table 4 and Table 6). The proposed simplified numerical model is suitable for practical purposes in industrial practice. It can be used to determine the response of new steel rope structures to their load or to design advanced rope structures in a more efficient manner.

## Figures and Tables

**Figure 1 materials-16-03756-f001:**
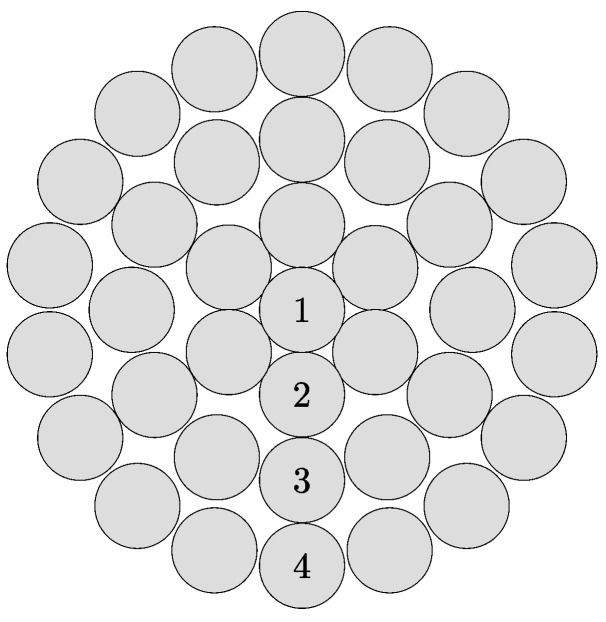
Cross–section of rope 1 × 37.

**Figure 2 materials-16-03756-f002:**
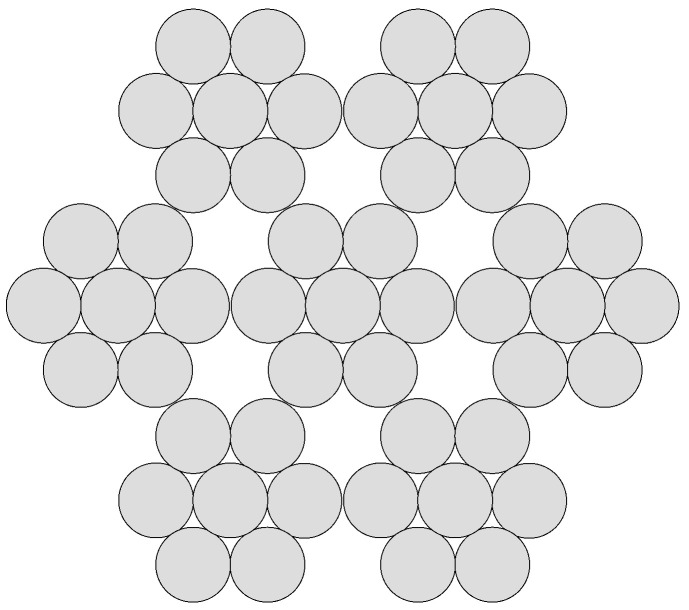
Cross–section of rope 6 × 7-WSC.

**Figure 3 materials-16-03756-f003:**
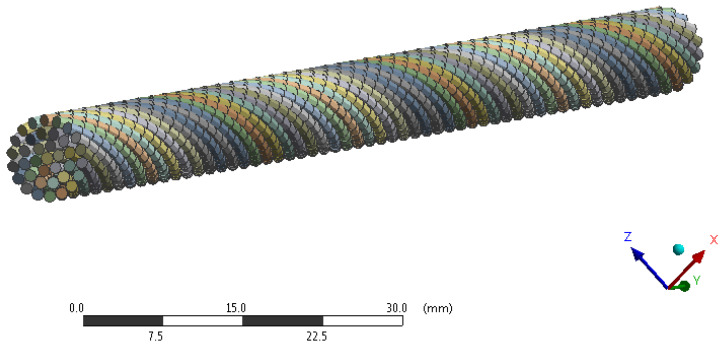
Finite element mesh of rope 1 × 37.

**Figure 4 materials-16-03756-f004:**
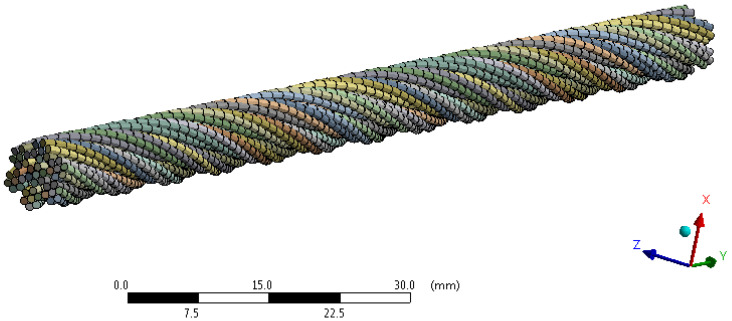
Finite element mesh of rope 6 × 7-WSC.

**Figure 5 materials-16-03756-f005:**
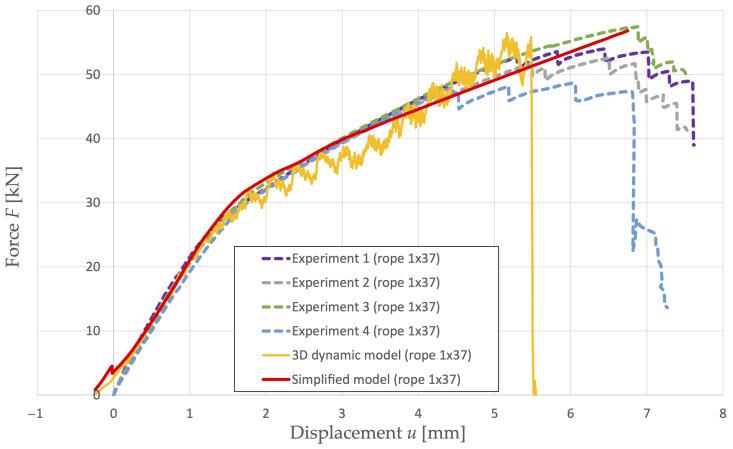
Comparison of 1 × 37 rope response in case of experiment and modeling [19].

**Figure 6 materials-16-03756-f006:**
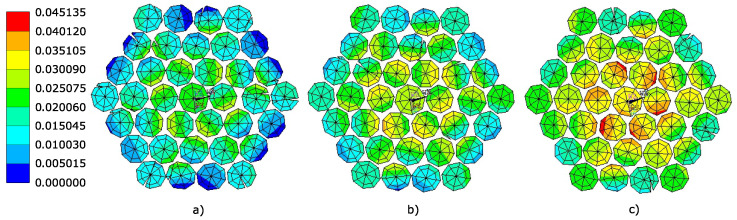
Equivalent plastic strain fields $\epsilon_{pl}$ [–] in simplified rope 1 × 37 cross-section corresponding to the axial force (**a**) $F_{c}$ = 45 kN, (**b**) $F_{c}$ = 50 kN, (**c**) $F_{c}$ = 55 kN.

**Figure 7 materials-16-03756-f007:**
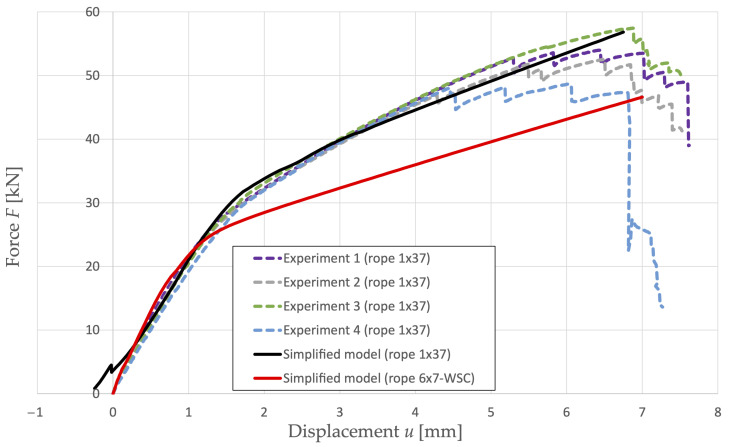
Rope 1 × 37 and rope 6 × 7-WSC response during tensile test.

**Figure 8 materials-16-03756-f008:**
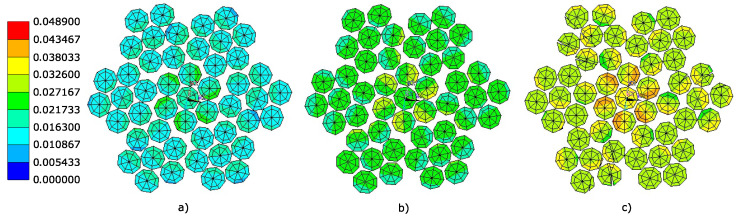
Equivalent plastic strain fields $\epsilon_{pl}$ [–] in simplified rope 6 × 7-WSC cross-section corresponding to the axial force (**a**) $F_{c}$ = 35 kN, (**b**) $F_{c}$ = 40 kN, (**c**) $F_{c}$ = 45 kN.

**Table 1 materials-16-03756-t001:** Wire diameters of individual rope layers of rope 1 × 37 [19].

Layer *i* [–]	1	2	3	4
Diameter $d_{d_{i}}$ [mm]	1.27	1.15	1.20	1.15

**Table 2 materials-16-03756-t002:** Finite element mesh statistics of both ropes.

Rope Construction	Nodes	Elements
1 × 37	7437	3700
6 × 7-WSC	9849	4900

**Table 3 materials-16-03756-t003:** Mechanical properties of rope 1 × 37.

Mechanical Property	Value
Modulus of elasticity *E*	200,000 [MPa]
Poisson ratio μ	0.30 [–]
Yield strength $\sigma_{y}$	1300 [MPa]
Tangent modulus $E_{t}$	22,200 [MPa]

**Table 4 materials-16-03756-t004:** Levels of load-bearing capacity and equivalent plastic strains of a simplified model of rope 1 × 37.

Load-Bearing Capacity $F_{c}$ [kN]	45	50	55
Equivalent plastic strain $\epsilon_{pl}$ [–]	3.19 × 10${}^{-2}$	3.93 × 10${}^{-2}$	4.51 × 10${}^{-2}$

**Table 5 materials-16-03756-t005:** Comparison of the load-bearing capacity of the steel rope and the failure of the first wire.

Experiment	1	2	3	4
First wire failure [kN]	53.6	51.8	57.5	48.0
Load-bearing capacity $F_{c}$ [kN]	54.0	52.5	57.5	48.7

**Table 6 materials-16-03756-t006:** Levels of load-bearing capacity and equivalent plastic strains of a simplified model of rope 6 × 7-WSC.

Load-Bearing Capacity $F_{c}$ [kN]	35	40	45
Equivalent plastic strain $\epsilon_{pl}$ [–]	3.30 × 10${}^{-2}$	4.10 × 10${}^{-2}$	4.89 × 10${}^{-2}$

## Data Availability

Data sharing not applicable.

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
