# Peer review of "Simplified Numerical Model for Determining Load-Bearing Capacity of Steel-Wire Ropes"

_materials, 2023, doi:10.3390/ma16103756_

Round 1
Reviewer 1 Report
1. The author should compare the stress-strain curve adopted by the finite element model with the mechanical properties tests; The reviewer personally believes that the error between the bilinear constitutive relation and the actual one will be relatively large.
2. The author defines the tangent friction coefficient of the contact pair. What are the properties of the contact pair in the normal direction?
3. The beam element used in the article is "BEAM188". What are the characteristics of this element, and how are its integral points arranged? Why choose this beam element?
4. In Figure 7, “step-change” phenomenon is observed in model 1×37 at the initial stage. Why this phenomenon is not observed in model 6×7-WSC?
5. “From the point of view of modelling, it is therefore assumed that the evaluated plastic strain ϵpl = 3.93E-02 belonging to the given load is the limit value for the given material model.” In the reviewer’s opinion, the ultimate strain of the material should only be obtained from the mechanical property test, rather than judged based on the finite element analysis, because the finite element analysis results are affected by many factors.
Reviewer 2 Report
Manuscript Number: materials-2264837
Full Title: Simplified Numerical Model for Determining Load-Bearing Capacity of Steel Wire Ropes
This manuscript aims at introducing simplified numerical model for steel wire ropes. What is presented is not even novel or complex to be subject of an scholarly article. This is just a simple numerical investigation which would be considered an engineering project rather that a research subject. Unfortunaltely, I cannot for its publication. My decision is based on the comments given hereafter.
Comments:
· Introduction is too generic, while it must be specific on the subject.
· The English needs revision:
o “… the rope can carries…”
o Punctuation rules should be obeyed.
o “Authors solve the task of….”
o The abstract should be rewritten.
o “…attaching to a load”
o “Wire steel rope” OR “steel wire rope”?
o “… to which this article follows.”
o …
· Regarding the results presented in Figures 5 or 7 it is clearly evident that the simulation is over-simplified and the minor failure event are not captured during simulation. Thus this would not be reliable modeling methodology.
· The details of plasticity model, failure model (if applied) and contact are not given.
· The results of 3D simulations are not given, so it cannot judged if the simplified 1D model works.
Reviewer 3 Report
Review of the manuscript
„Simplified Numerical Model for Determining Load-Bearing Capacity of Steel Wire Ropes” by Hroncek et al.
The presented manuscript describes an interesting approach to numerical modeling of the steel ropes widely used in the industry. The main advantage of the presented approach is to reduce the computational complexity and reduce the time needed for analysis. The manuscript is generally well structured except for the Conclusion chapter, which is too long for the volume of the presented work. Therefore, I am suggesting authors to focus on the main conclusions of the work and distribute the rest of the text to appropriate chapters.
The references should be updated with more current ones. Only five journal papers referenced are from the last five years of 24 references.
Another significant remark is heavy dependence on Ansys software. As an Ansys user, I am aware of its benefits, but the manuscript should give a more general description.
I do not feel qualified to judge English writing; however, I find the manuscript are used correct nomenclature and is generally understandable.
My specific remarks are:
- line 155: please describe BEAM188 element
- line 166: you are stating that you are using the material model with bilinear isotropic hardening. In table 3, you are providing some of the model's data. Please provide the rest of the data needed to reconstruct the material model. I would also prefer to see the stress-strain graph of the material used.
- How are you determining element failure? Is it strictly 1D stress or some 2D stress combination?
- line 180: the abbreviation GCGEN is not addressed
- line 181: please provide a description of element CONTA177
I suggest the authors make a major revision of the manuscript, incorporating all remarks above.
Reviewer 4 Report
I think this paper has been improved by the twice revisions and all concerns have been concerned.
For example, the background of the research field has been improved.
Besides, the numerical method has been clearly clarified and analyzed.
The research results are helpful for the actual engineering.
Author Response
Please find more details in the attached file.

Reviewer 5 Report
The authors should compare the results of their model with the existing models because simply claiming that one method for estimating load-bearing capacity is not sufficient. And it should be compared with the existing models and explain why it has a better performance or not.
Author Response

(The authors gave the same response as above.)

Round 2
Reviewer 3 Report
Thank you for accepting my suggestions.
The mauscript is ready to be published.
Author Response

(The authors gave the same response as above.)

Reviewer 5 Report
The respected authors have considered all reviewer's comment in the preparation of the revised version of the manuscript. This manuscript is recommended for publication at the present format.
Author Response
We appreciate your cooperation. We believe we have achieved the required manuscript quality.
Sincerely,
Dr. Pavel Marsalek, Ph.D